# Combined Membrane Dehumidification with Heat Exchangers Optimized Using CFD for High Efficiency HVAC Systems

**DOI:** 10.3390/membranes12040348

**Published:** 2022-03-22

**Authors:** Ajay Sekar Chandrasekaran, Andrew J. Fix, David M. Warsinger

**Affiliations:** Birck Nanotechnology Center, School of Mechanical Engineering, Purdue University, West Lafayette, IN 47907, USA

**Keywords:** membrane, dehumidification, energy exchanger, energy efficiency

## Abstract

Traditional air conditioning systems use a significant amount of energy on dehumidification by condensing water vapor out from the air. Membrane-based air conditioning systems help overcome this problem by avoiding condensation and treating the sensible and latent loads separately, using membranes that allow water vapor transport, but not air (nitrogen and oxygen). In this work, a computational fluid dynamics (CFD) model has been developed to predict the heat and mass transfer and concentration polarization performance of a novel active membrane-based energy exchanger (AMX). The novel design is the first of its kind to integrate both vapor removal via membranes and air cooling into one device. The heat transfer results from the CFD simulations are compared with common empirical correlations for similar geometries. The performance of the AMX is studied over a broad range of operating conditions using the compared CFD model. The results show that strong tradeoffs result in optimal values for the channel length (0.6–0.8 m) and the ratio of coil diameter to channel height (~0.5). Water vapor transport is best if the flow is just past the turbulence transition around 3000–5000 Reynolds number. These trends hold over a range of conditions and dimensions.

## 1. Introduction

### 1.1. Energy Consumption in Buildings

Both commercial and residential buildings consumed about 40% of the United States’ total primary energy in 2020 [1]. Of this, heating, ventilation and air conditioning (HVAC) accounts for 30% of the total building energy consumption [2]. In most current HVAC systems, dehumidifying the air by condensing out water vapor from air is energy intensive. This is estimated to consume 68% of the primary energy in commercial buildings [3]. Energy use in buildings has been increasing rapidly due to population growth, migration trends, and increased access to comfort control, which has resulted in a rapid increase in dehumidification loads. With this trend expected to rise on average by 5% per year due to climate change [4] the energy efficiency of buildings has become a major objective of recent energy policies [5]. Thus, the opportunities for improving the energy efficiency of building services are enormous, with potential to cut energy consumption by more than 30%, on average, by using more efficient technologies than those available on the market today [6].

### 1.2. Overview of Current HVAC Dehumidification

Traditionally, latent and sensible loads have been treated in a coupled manner such that the temperature of the cooling coils in an air conditioning system is set low enough to include both temperatures drop as well as condensation. Thus, the coils are maintained below the dew point of the air they are cooling, which is usually lower than is necessary to efficiently meet the sensible loads (desired temperature change). This coupling leads to large temperature gradients and large condensation heat loads in conventional air conditioning systems. Several research studies have also focused on improving the configurations, materials, and heat transfer performance of these conventional heat exchangers [7,8] but have begun to show diminishing returns [9,10]. Hence researchers have focused on studying alternative HVAC technologies like membrane-based HVAC systems. A study by the US Department of Energy showed that that these systems have the potential to save up to 2.3 Quads/year [2,11]. This study focuses on evaluating the performance of a novel configuration of a membrane energy exchanger. While prior work by our group has presented a thorough thermodynamic analysis of the technology, the CFD efforts in this work provide insight into the effect of physical parameters in the system, that are not included in generalized thermodynamic analyses.

#### 1.2.1. General Comparison of Alternative HVAC Technologies

Membranes have been used for many separation applications such as desalination [12,13], CO_2_ separation from flue gas (furnace exhaust gases), and dehumidification. A common air conditioning application is the membrane-based energy recovery ventilator (M-ERV) [14]. These devices passively transfer both sensible and latent energy between the incoming and exhaust air streams leaving an indoor space, which are separated by relatively non-selective membranes [15,16]. However, M-ERV’s can only provide partial dehumidification and cooling since they rely solely on temperature and humidity gradients between inlet and exhaust air streams to passively exchange energy, without any external work or heat input. Additionally, the relatively non-selective membranes in M-ERV’s can allow pollutants in from the indoor exhaust air stream to enter the fresh inlet air [17,18]. The technology analyzed in this work is fundamentally different from M-ERVs because it employs highly selective membranes and relies on significant pressure gradients but can provide complete cooling and dehumidification. This technology is also energy efficient in comparison to the liquid based desiccant air conditioning [19,20] which requires an energy intensive process for the desiccant regeneration [21]. Vapor selective membrane-based technologies are more energy efficient than conventional heat pump systems because they treat the latent and sensible loads independently with the help of the highly selective membrane, which allows the coiling coils to be optimized for sensible loads rather than the latent loads [22].

#### 1.2.2. Selective Membrane-Based Dehumidification

Vapor selective membrane-based dehumidification systems use a vacuum pump to create the driving force to mechanically separate the water vapor from an air stream. These systems use vapor/air selective membranes, which is a difficult separation to design for given the similarity in particle size compared to other membrane separation types. If this system is designed in an optimal configuration, it can be far more efficient than the condensation dehumidification used in conventional systems. Labban et al. (2017) and Fix et al. (2021) concluded that membrane-based cooling systems can achieve a system coefficient of performance (COP) two times higher than the current vapor compression cooling systems by analyzing several system designs with a first law-based modelling approach [22,23]. The most promising membrane technology in this study was the one which employed two vapor selective membrane modules [22]. This approach enables smaller pressure ratios across the vacuum pump/compressor, thus greatly reducing the energy input required compared to membrane dehumidification systems which reject the water vapor straight to atmospheric pressure [24]. For this reason, the membrane energy exchanger analysis in this work seeks to optimize the physical design parameters of a membrane module that would be part of the dual-module membrane dehumidification design and specifically focuses on the “Active Membrane Energy Exchanger” module concept, which includes cooling tubes within the membrane channel [25,26].

#### 1.2.3. Overview of CFD Modelling for Membrane Applications

Most previous works have focused on using CFD to study the membrane separation process of gasses involving CO_2_ or hydrogen [27,28] or to understand the system performance for dehumidification. For example, Bui et al. (2015) studied the isothermal (constant temperature) membrane dehumidification process using CFD to understand the tradeoff between membrane dehumidification and COP [29]. CFD models have also been widely used to study the effect of channel spacers and concentration polarization on heat and mass transfer performance in M-ERV’s to optimize the membrane dimensions [30,31]. In this study, we focus on CFD modelling of the heat and mass transfer of a membrane module that employs vapor selective membranes for dehumidification and cooling channels built into the module to provide simultaneous sensible cooling. The model presented herein can be used to understand important tradeoffs between the different performance metrics and geometric parameters of the system to guide real implementation.

#### 1.2.4. Scope and Novelty

Existing literature regarding selective membrane dehumidification generally falls into a few main categories: material development [32], thermodynamic modeling [33], basic experimental demonstration [34], and some CFD modeling [35], though much of the CFD modeling has focused on passive M-ERV’s, which are fundamentally different than the system studied in this work. In terms of existing literature on the combination of active heat exchanger and simultaneous membrane-based air dehumidification, only thermodynamic modeling work has been published by the authors [23,26,33]. These thermodynamic models employ assumptions and simplifications to understand system-level performance, independent of component sizing. This work provides the first physics-based analysis, using CFD, to provide a thorough understanding of the impact that sizing and geometry have on the performance of a combined heat exchanger and membrane dehumidification unit, with the goal of providing insight into optimal design parameters for real implementation.

## 2. Materials and Methods

### 2.1. System and Geometry Description

The membrane energy exchanger shown in Figure 1 is a non-isothermal system, which uses a vapor selective membrane to dehumidify the incoming air stream while simultaneously cooling the air. The selective membrane ensures that only water vapor passes through the membrane, facilitating the integration of the membrane and sensible cooling cycle within a single module [23].

Warm and humid air enters the feed side of a membrane module (Figure 1, left). A vacuum pump or compressor pulls the water vapor out of the incoming air and through the vapor selective membrane (Figure 1, middle). Ideally, this membrane will allow only water vapor transport, and not air, but in reality, the membrane would also allow a small amount of air to pass through due to its small permeance (K_air_ = 0.1 GPU) towards air. In the most efficient system using a vacuum pump, the extracted water vapor transfers to the feed side of an exhaust membrane module [36,37] (Figure 1, bottom) that is not shown here for brevity. The membrane module also cools the incoming air stream simultaneously via the cooling coils which could be the evaporator coils of a refrigeration vapor compression cycle. Since the dehumidification process lowers the dewpoint of the air stream along the length of the module, the sensible cooling can be designed such that the air temperature never drops below this dewpoint temperature, thus avoiding unwanted condensation. Perpendicular cooling channels were chosen in order to resemble traditional evaporator designs and to enable a cross-flow heat exchange configuration. Additionally, future work can investigate the viability of using the cooling channels as membrane supports or channel spacers.

Flat plate membrane systems are widely used due to their ease of manufacturing, and thin channels are commonly employed to minimize the convective mass transfer resistance. This design uses an in-line circular coil arrangement to facilitate a continuous temperature drop as the air flows across the channel [19]. The minimum diameter of the cooling coils is constrained by the minimum size currently available in the market. For simplicity, in the current system, the fluid in the cooling coils is not modelled explicitly, but rather, constant wall temperatures are assumed. The baseline dimensions of the system are chosen to reasonably compare with lab-scale membrane-based prototypes [38], like membrane distillation [38,39,40]. Figure 2 below shows the model geometry used in this study along with the initial base dimensions.

### 2.2. Modelling Methodology

A 2D CFD model was developed using STAR CCM+ to study the dehumidification performance, heat transfer, and mass transfer phenomena of the membrane energy exchanger. User defined equations are specified to define the mass transfer performance of the membrane, which is modelled as a thin permeable wall between the feed and permeate sides of the intake membrane module. The CFD results (temperature drop, condensation, and heat transfer coefficient) are compared with an analytical model and then used for parametrization The analytical model used for comparison here corresponds to external flow over a cylinder tube bank. The CFD model was developed according to the following assumptions:Humid air is considered to be an ideal gas mixture consisting of two components, water vapor and dry air.The fluid flow through the channel is considered to be steady and incompressible.The physical properties of the humid air are based on adiabatic mixing (i.e., based on the mass fraction of individual components of the mixture).The effect of temperature on the physical properties of the humid air mixture is negligible.Water vapor permeates through the membrane while there is negligible amount of air permeating through the membrane.The cooling coils are always maintained at a constant temperature.The vacuum pump always maintains a constant operating pressure gradient across the membrane.The fluid film formed on cooling coils during condensation (if any) offers a minimal resistance to the heat transfer occurring between the cooling coils and air stream in the channel.

A summary of the different test results obtained from CFD and the parameters that are kept constant or varied for each of these results are given in Table 1 below.

### 2.3. Governing Equations

#### 2.3.1. Mass and Momentum Conservation

The mass, momentum, energy, species transport, and condensation equations available in the CFD software are used, while the mass transfer across the membrane is specified by a user defined equation. The total mass conservation equation is used to ensure that the mass of vapor and air within the system [41] is conserved and is described by
(1)$$\nabla \cdot \left(\rho_{i}\vec{v}\right)=S_{m,i}\;$$
where $\rho_{i}$ is the density of each species i, and $\vec{v}$ is the velocity vector. $S_{m,i}$ is the rate of phase change of water due to condensation, $S_{m,i}$ is 0 for air since air does not undergo phase change), ∇ is the gradient of the given quantity. The mass conservation equation is solved for each species within the system. The momentum conservation equation is given by the following equation.
(2)$$\nabla \cdot \left(\rho \vec{v}\right)=-\nabla p+\nabla \overset{═}{\tau }\;$$
where ρ is the density of fluid mixture, ∇p is the pressure gradient responsible for the fluid flow in the system and $\overset{═}{\tau }$ is the stress tensor due to the shear stress. The negative sign indicates that the fluid flows from a high-pressure region to a low-pressure region. A single momentum equation is solved for all species present within the system, yielding the velocity of the air and the pressure drop in the channel.

#### 2.3.2. Energy Conservation

A single energy balance equation is solved for all species in the system and gives the temperature field. The energy conservation equation is defined according to Equation (3).
(3)$$\nabla \cdot \left(\rho \vec{v}c_{P}T\right)=\nabla k∆T+{\dot{m}}_{i}^{”}h_{\mathrm{fg}}\;$$

Here, *c_p_* is the specific heat of the fluid mixture at constant pressure, *k* is the thermal conductivity of the fluid mixture, ∆T is the temperature gradient in the system, ${\dot{m}}_{i}^{”}$ is the mass flux of condensation, and $h_{\mathrm{fg}}$ is the specific latent heat of condensation. The energy equation is used to solve for the heat transfer performance in the system.

#### 2.3.3. Species Transport

The species transport equation solves for concentrations of air and vapor species in the system based on Fick’s law of diffusion [41]. This is defined according to Equation (4).
(4)$$\nabla \cdot \left(\rho \vec{v}Y_{i}\right)=-\nabla \cdot \rho D_{i},\nabla Y_{i}\;$$

Here, $Y_{i}\;$ and *D*_i_ are the mass fraction and diffusivity, respectively, of each species (water vapor and air) within the system.

#### 2.3.4. Mass Transport through Membrane

The mass transport across the membrane interface is governed by a user-defined membrane permeance function [42] described as
(5)$$\frac{J}{A_{\mathrm{membrane}}}=K\;*\;\left(P_{\mathrm{vf}}-P_{\mathrm{vp}}\right)$$
where the left side represents the mass flux across the membrane interface, with J being the mass flow rate of each species through the membrane and *A*_membrane_ being the area of the membrane. *K* is the permeance of membrane, which is a measure of how well each species passes through the membrane. $P_{\mathrm{vf}}$ and $P_{\mathrm{vp}}$ are the partial pressure of each species on the feed and permeate sides of the membrane interfaces, respectively. Although, the mass transport equation across the membrane is a function of membrane properties like pore diameter, tortuosity, and porosity, a simple linear expression as a function of only vapor pressures and permeance has been used as a standard assumption [12].

#### 2.3.5. Condensation Mass Transfer

The condensation model is used to understand the rate of unwanted condensation by the cooling coils in the system. After optimizing the system dimensions to prevent condensation, the absence of condensation within the system can be verified by ensuring that the maximum value of relative humidity in the system does not exceed 1. The CFD model can predict RH values above one, which implies that a physical system would experience condensation. The condensation rate was determined according to Equation (6) [41].
(6)$${\dot{m}}_{i}=k_{c}*A_{\mathrm{coils}}*\rho_{H_{2}O}*\left(Y_{H_{2}O,\infty }-Y_{H_{2}O,w}\right)$$
where $\dot{m_{i}}$ is the condensation rate, $k_{c}$ is the mass transfer coefficient determined from the Schmidt number, $A_{\mathrm{coils}}$ is the cooling coil surface area, $\;\rho_{H_{2}O}$ is the density of water vapor,$\;Y_{H_{2}O,\infty }$ and $Y_{H_{2}O,w}$ are the mass fraction of vapor in the bulk air and on cooling coil walls, respectively. The equation is only activated when the vapor concentration at the wall interface reaches saturation conditions.

#### 2.3.6. Mesh Independence Study

A mesh independence study was conducted to ascertain that the CFD results do not depend on the mesh size. This study shows a mesh size of 0.007 m as the optimal size. The grid study meshes the region around the cooling coils using the prism layer meshing method to ensure a smooth mesh transition. The wake mesh setting within STAR CCM+ is also used, to prevent the formation of recirculation at the channel downstream. The mesh independence study plot is given in the Appendix A.

#### 2.3.7. Boundary and Operating Conditions

The dry air and water vapor mixture enters the feed channel at 27 °C with an absolute humidity of 0.025 (RH:70%) at the inlet. These conditions are chosen to represent a warm humid climate, such as that of India or the United Arab Emirates. The cooling coils were initially maintained at a constant temperature of 10 °C, comparable to current conventional systems, but this will be varied in part of the analysis. The pressure in the channel feed and permeate sides are 101.325 kPa (atmospheric pressure) and 1 kPa, respectively [43,44]. The removal of water vapor across the feed and permeate sides of the membrane is specified using values calculated from Equation (5). The membrane permeance in this study has a median value of 5000 GPU which matches several studies [45,46,47,48] and is assumed to have a high selectivity to water vapor (i.e., very little air passes through) [49,50]. Hence, the vapor and air mass fractions on the membrane permeate side are set as 0.99 and 0.01 for the initial conditions.

#### 2.3.8. Analytical Comparison

Comparison and verification of the CFD model is an important step towards ensuring that the model developed is accurate and reliable. The temperature, pressure drop, and heat transfer coefficient from the CFD model were compared with an analytical model of an external flow over a coil bank [51]. The mass transfer trends from the CFD match well with the results of a similar CFD model of a direct contact membrane distillation system [40]. The temperature drop across the coil bank is calculated using Equation SE1 from the Appendix A. Figure 3 shows the temperature drop comparison between the CFD and the analytical model, which yielded a maximum error of 2%. This error results from the difference in geometry and flow type between the CFD and analytical model. In the CFD model, the fluid flow is restricted by the walls of the channel. This increases the fluid’s velocity between the coil and top channel wall, which explains the quicker temperature drop shown in the CFD results compared to the analytical study. The average heat transfer coefficients from the analytical and CFD studies are 56 W/m^2^K and 60 W/m^2^K, respectively, with CFD showing a 7% higher value than the analytical value. The pressure drops from the analytical model, calculated using Equation SE3, and the CFD model are 27.8 Pa and 29 Pa, respectively, with the CFD model showing a 4.3% higher value than the analytical. The methods of calculation of these variables from the analytical model are described in the Appendix A.

## 3. Results and Discussion

The model was modified for a broad range of conditions and dimensions to obtain 50 data points for each of the contour plots between different design parameters of the membrane energy exchanger in the following sections.

### 3.1. Performance Study of Membrane Energy Exchanger: CFD Contour Plots

The temperature of the fluid decreases along the channel length, and an optimum number of cooling coils needs to be provided to handle the required sensible load. The cooling occurs abruptly at each cooling coil, and the system is operating in a slightly turbulent regime, as seen in the temperature profile. Narrow channel height and longer channel length help reduce the effect of convective mass transfer resistance so that the membrane can sufficiently dehumidify the entire volume of air before it reaches the MHX outlet. The coils are placed far enough apart from each other such that the temperature of the air (sensible cooling) never drops below the dewpoint temperature (which is dependent on the dehumidification rate).

CFD contours plot of the temperature, relative humidity, and absolute humidity are shown in Figure 4.

### 3.2. Concentration Polarization Dependence on Reynolds Number and Membrane Permeance

Concentration polarization in this study is the ratio of vapor concentration at the membrane surface to that in the bulk. Concentration polarization is an important performance characteristic that can impair the mass transfer of membrane-based systems. Here, concentration polarization occurs when the humidity level at the membrane interface is substantially lower than the humidity level in the bulk air stream. This phenomenon leads to lower dehumidification rates than would otherwise be expected when using the bulk air stream humidity to calculate dehumidification rates. Comparing this with the Reynolds number and membrane permeance (Figure 5) can help optimize the membrane mass transfer effectiveness.

At a smaller Reynolds number and membrane permeance, there is very little concentration polarization (bottom left region). Increasing the Reynolds number increases the mixing within the channel, decreasing the effect of concentration polarization. Higher membrane permeance leads to a higher concentration polarization as the membrane causes a greater deficiency of vapor at the membrane surface (bottom right region). Membrane permeance in the range of 4000–6000 GPU is generally found to be the optimal value to avoid effects of concentration polarization.

### 3.3. Effect of Channel Height and Coil Diameter on the Pressure Drop

The effect of diameter to channel height ratio (d/h ratio) and Reynolds number on the pressure drop is depicted in Figure 6. The pressure drop in the channel increases with a greater coil diameter to channel height ratio (d/h) and by increasing the Reynolds number. Increasing the d/h ratio has two effects: first, the cooling coil occupies a greater portion of the flow area, greatly increasing the frictional losses and second, the air is forced to move at increased velocities around the coil due to the reduced flow area. This locally elevated velocity implies a locally elevated Reynolds number, further exacerbating the pressure drop. When the average Reynolds number in the channel is in the laminar region (less than 2000), increasing the d/h ratio from 0.1 to 0.9 increases the pressure drop, on average, by 100%. As the flow becomes more turbulent (near Re = 4000), increasing the d/h ratio from 0.1 to 0.9 increases the corresponding pressure drop on average by 450%. From the viewpoint of aiming to minimize the pressure drop in the air channel, a smaller d/h ratio with laminar flow is desirable. However, this would lead to poor cooling performance as well, where larger d/h and Reynolds number will be beneficial. So, to balance this tradeoff, a module designed with a d/h ratio of approximately 0.5 and an operating Reynolds number in the range of 3000–5000 will maintain pressure drops comparable to conventional air conditioning systems while still providing reasonable cooling capability. The effect of coil diameter to channel height ratio on the heat transfer is discussed in detail in Appendix A.

### 3.4. Tradeoff between Channel Length and Reynolds Number on Membrane Area

Another important tradeoff for high performing systems revolves around the membrane area. Comparing membrane area with Reynolds number and channel length (Figure 7) is the broadest way to see these key design tradeoffs. In this analysis, the water vapor removal rate was set constant, and the output membrane area was normalized (divided by) the water vapor removal rate to provide results in a more generalized manner. So, for a very short channel, the membrane must be very wide to provide sufficient mass transfer area. Additionally, a higher Reynolds number will lead to greater mixing of the flow, enhancing the mass transfer. To further elucidate these tradeoffs, we provide an example. For a given a channel length of 0.3m, increasing the Reynolds number from 1000 to 4000 decreases the membrane area required by 50%, due to the enhanced mixing. Further increasing the Reynolds number beyond the turbulent regime yields minimal benefits (Figure 7, top left). As the channel length is increased, the width of the membrane can become smaller while still meeting the constant vapor removal. Increasing the channel length beyond a threshold value does not yield any additional benefits (right portion of Figure 7). For example, increasing the channel length from 0.2 to 0.6m decreases the membrane area by 68% but increasing the channel length beyond 0.6m yields diminishing benefits. This stems back to the concept of concentration polarization. Increasing the channel length means that a greater portion of the flow will experience significant concentration polarization. Since concentration polarization reduces the dehumidification driving force, the total membrane area increases at longer channel lengths to meet the set vapor removal rate. The optimal length will vary for different configurations and constraints, but these results highlight how the optimal channel length can be identified. Most commercial membrane-based systems have a channel length on the order of 1–2 m [12].

### 3.5. Membrane Permeance vs. Membrane Area Tradeoff

The required membrane area is also a strong function of the membrane permeance to water vapor and desired humidity reduction of the air stream (Figure 8). For smaller amounts of humidity reduction around 0.001–0.0015 kg moisture/kg dry air, increasing the membrane permeance from 2000 GPU to 10,000 GPU yields a negligible reduction in the membrane area required to meet the set vapor removal rate. At higher humidity reductions around 0.002–0.035 kg/kg, increasing the membrane permeance from 2000 GPU to 6000 GPU decreases the required membrane area by an average of 65%. Increasing the membrane permeance further does not result in significant membrane area reductions. Hence, a membrane permeance around 5000–6000 GPU can provide near-optimal dehumidification, avoids significant concentration polarization (see Figure 5), and several materials already exist with these permeance values [32,48]. Additionally, it was identified that the membrane area required for dehumidification is 3 times more than the area of the cooling coils required to provide the sensible cooling.

### 3.6. Effect of Reynolds Number on Number of Cooling Coils

The number of cooling coils in the channel is dependent on the sensible cooling requirement. The relationship between the required number of cooling coils, Reynolds number, and temperature is shown in Figure 9. To achieve cooler air in extremely hot climates (a temperature difference of 20 °C for example), a large number of coils (20) is required when the flow is laminar. Increasing the Reynolds number from laminar to the transition regime (Re~2300) reduces the number of coils required by approximately 55% and increasing the Reynolds number from laminar (Figure 9 bottom) to the turbulent regime (Figure 9 top) reduces the number of coils required by approximately 78%. For the given geometry, it can be seen that the required number of cooling coils is a much stronger function of the Reynolds number than of the temperature drop.

### 3.7. The Effect of Membrane Permeance on Horizontal Coil Spacing

The membrane permeance to water vapor, coil temperature, and coil spacing are all tied to one another in terms of avoiding unwanted condensation in the system. By varying the temperature of the cooling coils (or more specifically, the temperature difference between the cooling coils and the inlet air dewpoint) and the membrane permeance (which dictates the dehumidification rate), the minimum coil spacing that avoids condensation can be determined (Figure 10). A larger coil spacing leads to more gradual temperature drop, ensuring the temperature of the air never drops below the dewpoint of the air. The y axis in Figure 10 (∆T) represents the difference between the dew point temperature for the given inlet conditions, and varying coil temperatures. Increasing membrane permeance from 2000 GPU to 6000 GPU for the same temperature difference reduced the coil spacing by 60% as expected. A similar explanation can be given for maintaining coil temperature closer to dew point for different membrane permeance values.

While this plot shows uniform horizontal coil spacing required to avoid condensation, the total channel length can be minimized by using a variable coil spacing along the length. Since the air is dehumidified along the length, its dewpoint will decrease. Thus, the coils closer to the inlet of the channel need to be placed further downstream to avoid condensation and the coil spacing thereafter can become progressively smaller as the dewpoint of the air is substantially reduced by the membrane dehumidification.

## 4. Conclusions

A CFD model was built to develop contour plots which could be used to understand the tradeoffs between different design parameters of a membrane energy exchanger. The important conclusions from the analyses are given below.

The membrane area required for dehumidification and the effects of concentration polarization are minimized with a turbulent Reynolds number and an optimal channel length. However, operating at high turbulence regimes can also result in higher pressure drops. In general, a Reynolds number in the range of 3000–5000 was found to reasonably avoid significant concentration polarization effects, maintain manageable pressure drops, and achieve minimum membrane area requirements. These values are subject to change for different configurations.Coil diameter to channel height ratios (d/h) in the range of 0.1–0.5. corresponded to pressure drops in the range of 50–320 Pa depending on the Reynolds number. Operating in the Reynolds number range suggested in the previous conclusion point with a d/h ratio of 0.5 maintains reasonable pressure drops while also enabling the design to provide sufficient air cooling.A variable horizontal coil spacing can help further minimize the channel length while still avoiding condensation, however variable coil spacing was not explicitly investigated in this work.The optimal membrane permeance value was found to be in the range of 5000–6000 GPU which would give manageable membrane area and avoid the effects of concentration polarization. Increasing the membrane permeance further does not yield major benefits for all cases consideredThe area of membrane required for mass transfer is at-least 3 times greater than the area of the cooling coils required for sensible cooling. This value is very specific to the given geometry and assumed operating conditions and is subject to change in a practical application. But it highlights the need to optimize these systems for both heat and mass transfer.The ideal length of the channel for the given configuration is in the range of 0.6–0.8 m. Increasing the channel length beyond this value does not yield significant benefit. Furthermore, the ideal cooling coil diameter-to-height ratio was 0.5 operating.

This work provides detailed insight into the optimal design parameters for system that combines heat exchange and membrane dehumidification into one process. While it would be ideal to compare the model against experimental data, no such experimental data exists in the literature for this type of system. However, the authors are currently developing an experimental test bench and plan to publish experimental results in the near future, which will complement the understanding provided by this work. A second generation of the CFD model presented in this work is under development for direct comparison and validation with the experimental results.

Future development work should focus on studying the reliability of these systems over long term operation. Fouling of the membranes is another key area of research that has gained minimal attention for dehumidification applications but is a major concern in other membrane technologies [12]. Furthermore, efforts on developing manufacturing capabilities for producing optimized systems is necessary for the full realization of the technology. Lastly, more work is needed to scale these systems, both to large scales systems for buildings and small, lightweight sizes for applications like electric vehicles [52].

## Figures and Tables

**Figure 1 membranes-12-00348-f001:**
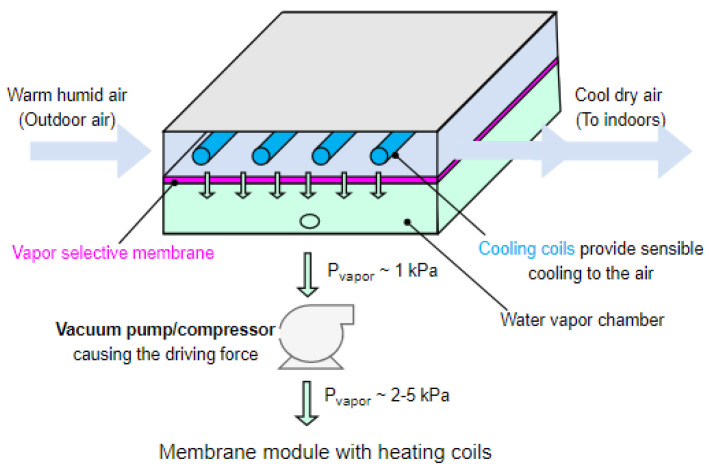
Membrane energy exchanger design. This design simultaneously cools the warm humid air while dehumidifying it with the membrane, helping to avoid the high energy consumption associated with the condensation of water vapor.

**Figure 2 membranes-12-00348-f002:**
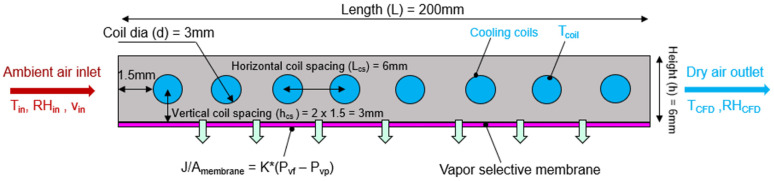
Baseline dimensions of the membrane energy exchanger 2D CFD study. Baseline dimensions are used to initially study membrane energy exchanger performance and are optimized further by CFD parametric sweep results. The boundary conditions used for the CFD model are also shown. The inlet boundary conditions are chosen to represent a warm humid climate at the inlet and typical HVAC operating conditions at the outlet. The inlet boundary condition is maintained at 27 °C, 70% RH (relative humidity) for most cases.

**Figure 3 membranes-12-00348-f003:**
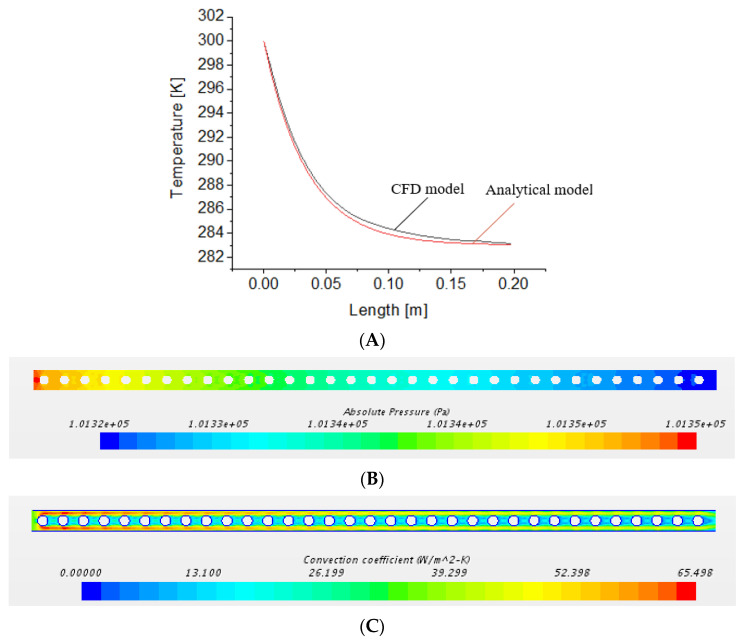
(**A**) Comparison of the CFD model with the analytical method, modelling cooling of air by coils perpendicular to the flow direction. The graph shows the temperature drop along channel length matches well between the analytical and CFD models. (**B**) The result shows the pressure drop along channel length from CFD. The pressure drop is uniform across channel width and decreases gradually along the channel length. The analytical model used for pressure drop calculation can be found in the Appendix A. (**C**) The result shows the convection coefficient along channel length from CFD. Higher convection coefficient occurs around vicinity of coils due to higher velocities in these regions. The analytical model used for convection coefficient calculation can be found in the Appendix A.

**Figure 4 membranes-12-00348-f004:**
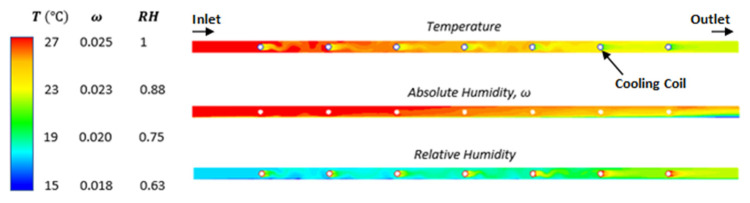
CFD contour results for temperature (top), specific humidity (middle) and RH (bottom). L × W × H = 0.6 m × 0.006 m × 1 m v_in_ = 5m/s, T_in_ = 27 °C, RH_in_ = 70%, T_coil_ = 10 °C, B = 3500 GPU. Turbulent flow (Re = 4500) improves heat and mass transfer performance which is particularly visible in the temperature contour.

**Figure 5 membranes-12-00348-f005:**
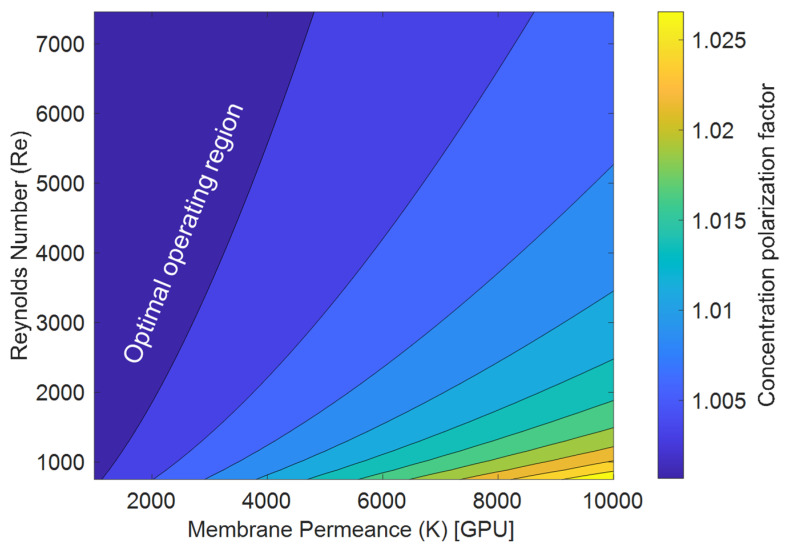
Effect of Reynolds number and membrane permeance on concentration polarization, calculated by the ratio of air concentration at membrane surface to the air concentration in the bulk of the channel. Operating the membrane energy exchanger in the turbulent regime helps avoid concentration polarization (top left region). T_in_ = 27 °C, RH_in_ = 70%, T_coil_ = 10 °C, d = 3 mm with L × H × W as 0.8 m × 0.006 m × 0.3 m, N_coils_ = 8, h_cs_ = 0.006 m and L_cs_ = 0.08 m.

**Figure 6 membranes-12-00348-f006:**
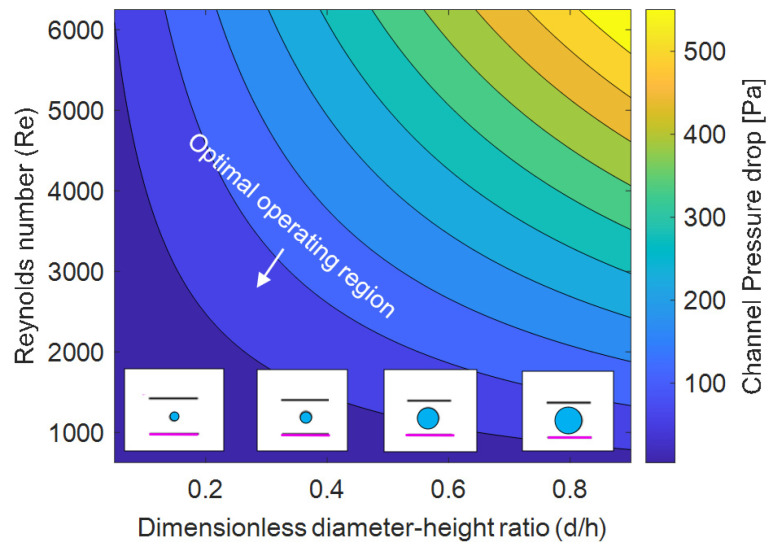
Pressure drop analysis. There are substantial changes in pressure drop once the flow is turbulent (Re > 2300) and larger diameters also cause substantial pressure penalties. T_in_ = 27 °C, RH_in_ = 70%, T_coil_ = 10 °C, L × H × W = 0.8 m × 0.006 m × 0.3 m, N_coils_ = 8, h_cs_ = 0.006 m, L_cs_ = 0.08 m, B = 5000 GPU.

**Figure 7 membranes-12-00348-f007:**
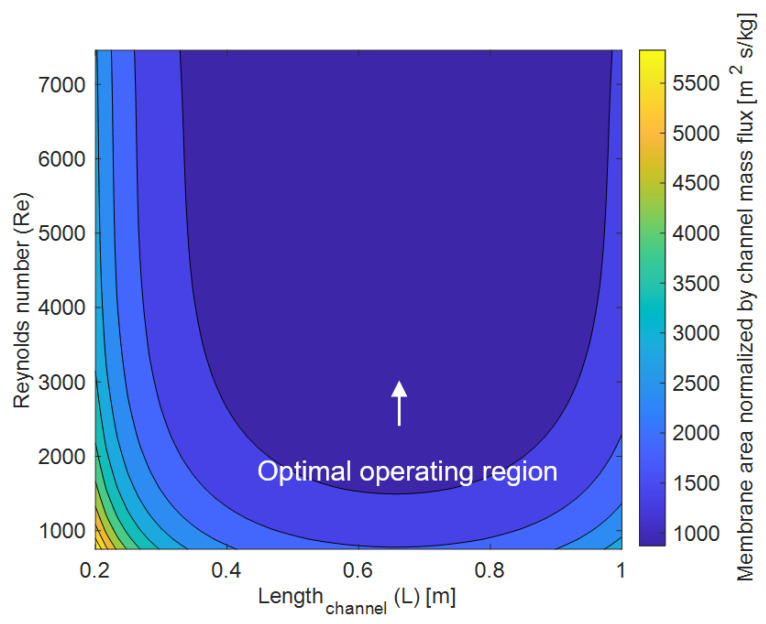
Membrane surface area requirements. T_in_ = 27 °C, RH_in_ = 70%, T_coil_ = 10 °C, H = 0.006 m, N_coils_ = 8, d = 0.003 m, h_cs_ = 0.006 m. B = 5000 GPU while the horizontal coil spacing is varied to accommodate 8 coils depending on the channel length. W is varied to produce J/A_membrane_ = 0.00005 kg/s.

**Figure 8 membranes-12-00348-f008:**
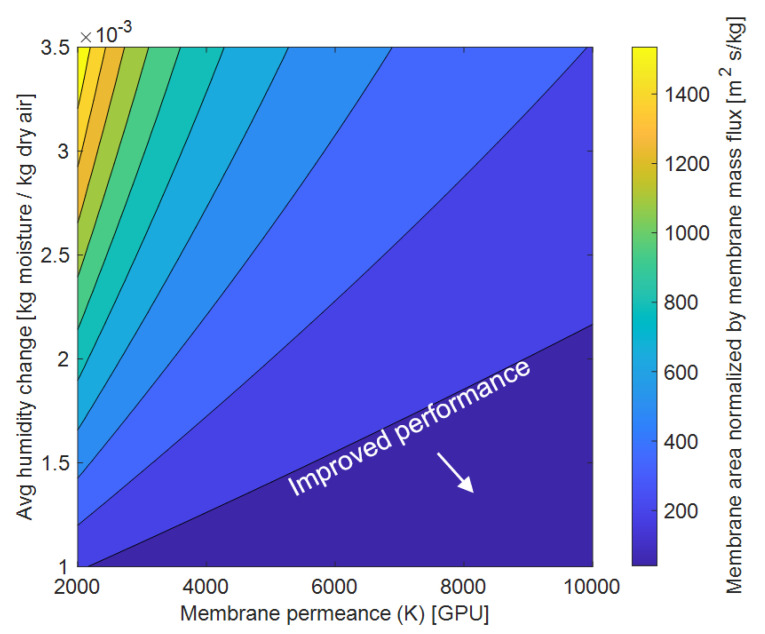
Impact of membrane permeance and humidity reduction on the normalize membrane area. Increasing permeance yields diminishing returns especially for smaller dehumidification needs (bottom right region). These conditions are T_coil_ = 10 °C, H = 0.006 m, N_coils_ = 8, and h_cs_ = 0.006 m.

**Figure 9 membranes-12-00348-f009:**
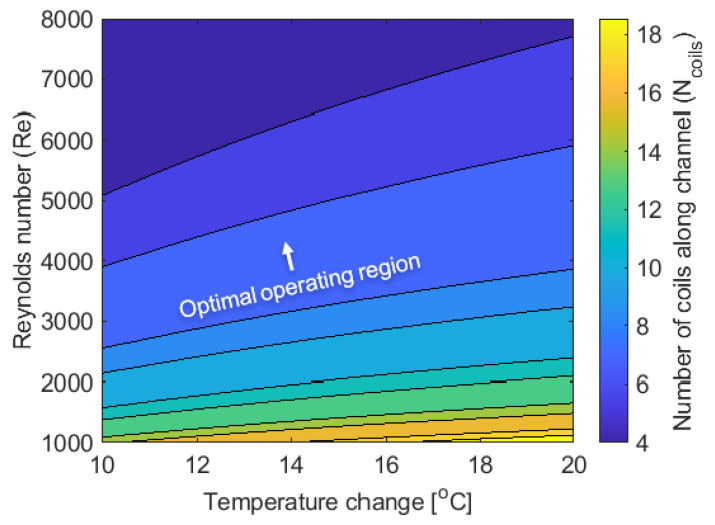
Number of coils required for cooling of air with respect to Reynolds number. T_in_ = 27 °C, RH_in_ = 70%, with H × W as 0.006 m × 1 m, h_cs_ = 0.006 m. B = 5000 GPU. Transitioning from laminar to turbulent flow causes a substantial decrease in the required number of coils; Further increases of Reynolds number yield diminishing returns.

**Figure 10 membranes-12-00348-f010:**
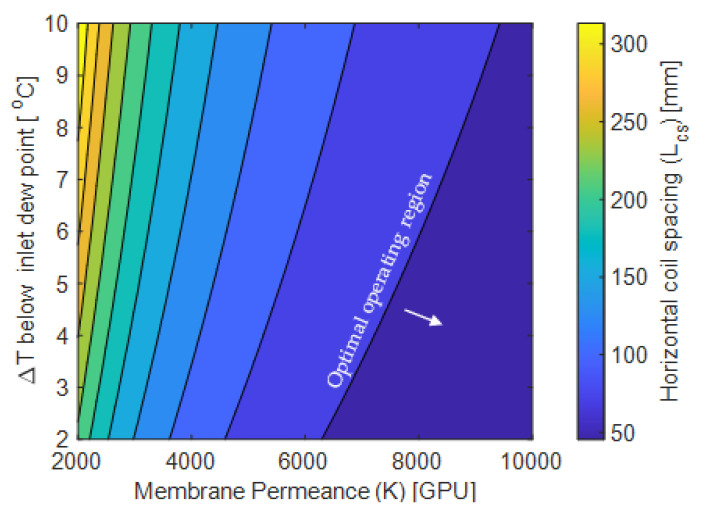
Horizontal coil spacing optimization. T_in_ = 27 °C, RH_in_ = 70%, with L × H × W as 0.8 m × 0.006 m × 1m, h_cs_ = 0.006 m, Re = 4000. The number of cooling coils were decided based on horizontal coil spacing and target temperature to be achieved at the channel outlet.

**Table 1 membranes-12-00348-t001:** Summary of test results from CFD with constant and variable parameters used for obtaining each result.

Section	Study Name	Constant Parameters	Variable Parameters
Section 3.1	Concentration polarization dependence on Reynolds Number and Membrane Permeance	Channel dimensions (L, W, H), N_coils_, L_cs_, h_cs_, RH_in_, *T*_in_, *T*_coil_, *A*_membrane_, d.	Membrane permeance (*K*), Reynolds number (Re, by varying inlet velocity).
Section 3.2	Effect of channel height and coil diameter on the pressure drop	L, W, H, N_coils_, L_cs_, h_cs_, RH_in_, *T*_in_, *T*_coil,_ d, *A*_membrane_, *K*.	Re (by varying inlet velocity), coil diameter
Section 3.1	Tradeoff between channel length and Reynolds number on membrane area	H, L_cs_, K, T_in,_ RH_in_, *T*_coil_, membrane mass flux, N_coils_.	Re (by varying inlet velocity), L, *A*_membrane_, h_cs,_ channel width(W).
Section 3.1	Membrane permeance vs. membrane area tradeoff	H, T_in_, N_coils_, *T*_coil_, h_cs._	K, L_cs_, L, W, A_membrane_, membrane mass flow rate ($\dot{m}$_membrane_)
Section 3.1	Effect of Reynolds number on number of cooling coils	L, W, H, h_cs_, *K*, *T*_in_, RH_in_.	Re (by varying inlet velocity), N_coils_, L_cs_, *T*_coil_
Section 3.1	The effect of membrane permeance on horizontal coil spacing	L, W, H, Re, h_cs_, *T_i_*_n_, RH_in_.	K, L_cs_, *T*_coil_

## Data Availability

No experimental data was collected as part of this work. All plots were generated from simulated values using the modeling approach described herein.

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
