# Peer review of "Combined Membrane Dehumidification with Heat Exchangers Optimized Using CFD for High Efficiency HVAC Systems"

_membranes, 2022, doi:10.3390/membranes12040348_

Round 1

Reviewer 1 Report

  • Acronyms and abbreviations need to be explained the first that they appear
  • Bulk citations (e.g. [7–10]) need to be avoided in the whole manuscript
  • In Section 1 the Authors need to describe the current status on the investigated topic and then clearly state what knowledge gap their work will fill COMPARED TO the current status on the investigated topic
  • Subsection 1.3 should be moved to Section 2
  • In the whole manuscript: Results of CFD analysis can be validated ONLY against field measurements/experimental data
  • Please, summarize and quantify the main findings as well as present the major conclusions in Section 5. Also, in this Section please summarize the limitations and the necessary future developments of your work
  • Nomenclature is missing

Reviewer 2 Report

The authors proposed an interesting concept " membrane-based energy exchanger " and did CFD simulation on its performance. The report is compact, accurate and well written for the expected audience. It is also very well oriented to technology, fitting well in the scope of the journal. There are some concerns to be addressed.

1. The physical model used for verification needs to be explained in detail. The average heat transfer coefficient and pressure drop verification also need to be shown in the figure.

2. The definitions of Concentration polarization and membrane area normalization in section 4.1 and 4.3, respectively, need to be described.

3. This research will be helpful for the dehumidification of the HVAC system of electric vehicles. It is recommended to refer to the following paper : https://doi.org/10.3390/en14010046

Round 2

Reviewer 1 Report

  • Subsection 1.2.3: "...involving CO2 (carbon dioxide) or hydrogen..."

There is no need to explain what CO2 is

  • The word validation/validate needs to be replaced with comparison/compare in the whole manuscript. Results of CFD analysis can be validated ONLY against field measurements/experimental data
  • The symbol indicating membrane mass flow rate needs an overdot in the whole manuscript
  • c_p indicates specific heat at constant pressure in the whole manuscript
  • m_i needs an overdot in the whole manuscript
  • h_fg indicates specific latent heat of condensation in the whole manuscript
  • kJ rather than KJ in the whole manuscript

Reviewer 2 Report

I see that the authors have carefully considered all the referees' remarks and have made all the appropriate changes. These changes have improved the paper significantly. I would like to recommend this paper for publication.
